

# Estimating sand bed load in rivers by tracking dunes: a comparison of methods based on bed elevation time series

**Kate C. P. Leary**[1,2] **and Daniel Buscombe**[3]

[1]Department of Geography, University of California Santa Barbara, Santa Barbara, CA, USA
[2]Department of Earth and Environmental Sciences, New Mexico Tech, Socorro, NM, USA
[3]School of Earth & Sustainability, Northern Arizona University, Flagstaff, AZ, USA

**Correspondence:** Kate C. P. Leary (learykcp@ucsb.edu)

**Abstract.** Quantifying bed-load transport is paramount to the effective management of rivers with sand or gravel-dominated bed material. However, a practical and scalable field methodology for reliably estimating bed load remains elusive. A popular approach involves calculating transport from the geometry and celerity of migrating bedforms, extracted from time series of bed elevation profiles (BEPs) acquired using echo sounders. There are various echo sounder sampling methodologies to extract bed elevation profiles. Using two sets of repeat multibeam sonar surveys with high spatiotemporal resolution and coverage, we compute bed load using three field techniques (one actual and two simulated) for acquiring BEPs: repeat multibeam, single-beam, and multiple single-beam sonar. Significant differences in flux arise between repeat multibeam and single-beam sonar. Multibeam and multiple single-beam sonar systems can potentially yield comparable results, but the latter relies on knowledge of bedform geometries and flow that collectively inform optimal beam spacing and sampling rate. These results serve as a guide for design of optimal sampling and for comparing transport estimates from different sonar configurations.

## 1 Introduction

Bed load is usually a significant proportion of total transported load in rivers with sand and/or gravel-dominated bed material, and the relative importance of suspended load and bed load often changes with flow and the location within the channel (e.g., Gomez, 1991). Whereas instrumentation and protocols for sampling suspended sediment loads are relatively well established (e.g., Nolan et al., 2005; Wren et al., 2000; Edwards and Glysson, 1999), reliable estimates of bed load are more difficult to obtain because bed load in transport is difficult to sample directly (e.g., Emmett, 1980; Gomez, 1991), define (e.g., Church, 2006; Yang, 1986), or estimate with empirical formulas (e.g., van Rijn, 1984; Martin and Church, 2000). Therefore, the effectiveness of sediment management in river systems is often predicated on the accuracy and representativeness of available bed-load measurements.

Reliable estimates of bed-load transport can be produced from application of the Exner equation (Simons et al., 1965; Engel and Lau, 1980) to time series of bed elevation profiles (BEPs) (Simons et al., 1965; Van Den Berg, 1987; Dinehart, 2002; Villard and Church, 2003; Wilbers and Ten Brinke, 2003; Claude et al., 2012; Guala et al., 2014) acquired with an echo sounder. Simons et al. (1965) showed that bed-load flux can be estimated by tracking the average celerity, $V_c$, of the downstream migration of dunes with a known average height, $H$, and average length, $\lambda$. These variables are averaged over a field of dunes to satisfy the necessary assumptions that suspended sediment load, $q_s$, is in equilibrium ($dq_s/dx = 0$), and with continuity of mass ($dq_b/dx + d\eta/dt = 0$), where $x$ and $\eta$ are downstream distance and bed elevation, respectively (Simons et al., 1965). The Simons et al. (1965) approach therefore quantifies only the first-order bed-load flux due to dune translation, not accounting for any

exchanges in bed material load between suspended and bed-load fractions that deform the dune and may contribute to net transport (McElroy and Mohrig, 2009).

Complicating matters, however, is the inherent variability of bedform size and shape in natural river systems (Bradley and Venditti, 2017). Sediment and water discharge conditions vary continuously in natural rivers causing bed morphology to be out of equilibrium with prevailing flow conditions. Numerous field studies suggest that bedform disequilibrium is likely a standard phenomenon in natural river systems (e.g., Frings and Kleinhans, 2008; Julien et al., 2002; Wilbers and Ten Brinke, 2003; Ten Brinke et al., 1999). Even with these complications, bed-load flux estimated from translating dunes remains one of the most accurate bed-load estimate techniques for sand-bedded rivers (Nittrouer et al., 2008; Wilbers and Ten Brinke, 2003; Holmes and Holmes, 2010) and as such bed elevation measurements are of great importance.

In practice, bed elevation profiles (BEPs) might be acquired in three ways using echo sounders: repeat multibeam, single-beam, and multiple single-beam sonar (Fig. 1a). Repeat multibeam surveys measure a spatially extensive three-dimensional bed, $\eta(x, y, t)$, from a moving vessel, from which it is possible to independently and simultaneously estimate $V_c$, $H$, and $\lambda$. Single-beam systems measure a one-dimensional bed, $\eta(t)$, at a single $(x, y)$ location using a stationary (fixed reference frame) sonar. Single-beam echo sounders have also been used to acquire data from a moving vessel (specifically for longitudinal profiles in shallow environments). The study presented herein does not address BEPs and resulting flux estimates calculated from moving vessel single-beam methods. Lastly, multiple single-beam surveys measure a spatially limited three-dimensional bed, $\eta(x, y, t)$, at a few $(x, y)$ locations using stationary sonar. All three sampling methodologies have been and are currently employed to collect bed elevation data in experimental (Blom et al., 2003; van der Mark et al., 2008; Guala et al., 2014; Curran et al., 2015), natural fluvial (Simons et al., 1965; Ten Brinke et al., 1999; Ashworth et al., 2000; Julien et al., 2002; Parsons et al., 2005; Gaeuman and Jacobson, 2007; Nittrouer et al., 2008; Claude et al., 2012; Shugar et al., 2010; Rodrigues et al., 2015; Wintenberger et al., 2015; Huizinga, 2016; Kaplinski et al., 2017; Buscombe et al., 2017; Hackney et al., 2018), and estuarine/coastal environments (Villard and Church, 2003; Hoekstra et al., 2004; Schmitt and Mitchell, 2014).

There are practical benefits and drawbacks to each data collection method. Repeat multibeam is spatially extensive, but relatively expensive, only practical in relatively deep, safely navigable rivers, and limited in temporal coverage. Therefore, the use of in situ stationary echo sounders is becoming an increasingly popular alternative (Gray et al., 2010). These systems are especially useful in shallow water, are less expensive than multibeam systems, and generate longer time series (e.g., Moulton et al., 2014). However,

only measuring the bed elevation at a single location means it is not possible to resolve $V_c$, $H$, and $\lambda$ simultaneously and, since space and time are not linearly substitutable (Guala et al., 2014), not trivial to estimate $\lambda$ from $V_c$ or vice versa. This has implications for bed-load estimates that are explored in this paper.

Since different methodologies may be employed to collect BEPs, and thus calculate bed-load transport rates, it is important that the resulting bed-load flux estimates from each method are compared to test for consistency. Do differences in sampling methodology cause systematic differences in BEPs and bed-load flux estimates? Are different sampling methods equally able to measure/account for changes in bedform size and shape with changing flow conditions? If not, how do changes in bedform size affect the bed elevation measurements made by different types of echo sounders?

Presently, it is unclear how differences in bed elevation data acquired with different methodologies translate to the fidelity with which dune migration is captured and finally to bed-load transport estimates. Furthermore, multibeam, single-beam, and multiple single-beam datasets do not generally exist in the same location and time, making direct comparison of these datasets difficult. In order to examine these issues, we establish a virtual experiment using a repeat multibeam dataset to directly compare bed-load transport estimates calculated from BEPs extracted to mimic the three different field survey methodologies outlined above.

## 2 Methods

### 2.1 Study area and survey data

We use an extensive repeat multibeam dataset consisting of bed elevation from a large area of migrating dunes at high spatiotemporal resolution. Data come from an approx. 300 m long by 40 m wide reach upstream of the Diamond Creek USGS gage site on the Colorado River in Grand Canyon National Park (Fig. 1b, c), where flows are regulated by releases through Glen Canyon Dam 385 km upstream. Flow depths at the downstream extent ranged from 6.8 m on the lowest flow of 310 m³ s⁻¹, and 7.9 m on the highest flow of 590 m³ s⁻¹, although there are deeper holes within the surveyed reach. We simulate data from simultaneous single-beam and multiple single-beam deployments by extracting time series of bed elevations from the repeat multibeam datasets (Fig. 1d). This "virtual echo sounder" experiment allows us to directly compare flux estimates from all three methodologies. We assess the relative accuracy of the single-beam and multiple single-beam techniques at estimating bed-load transport compared to repeat multibeam-derived bed load and suggest practical guidelines for developing sampling and processing protocols that maximize accuracy.

Repeat multibeam surveys were collected at two different discharges (Fig. 2a) from just upstream of the Diamond Creek USGS gage site (Fig. 1c). All bathymetric data

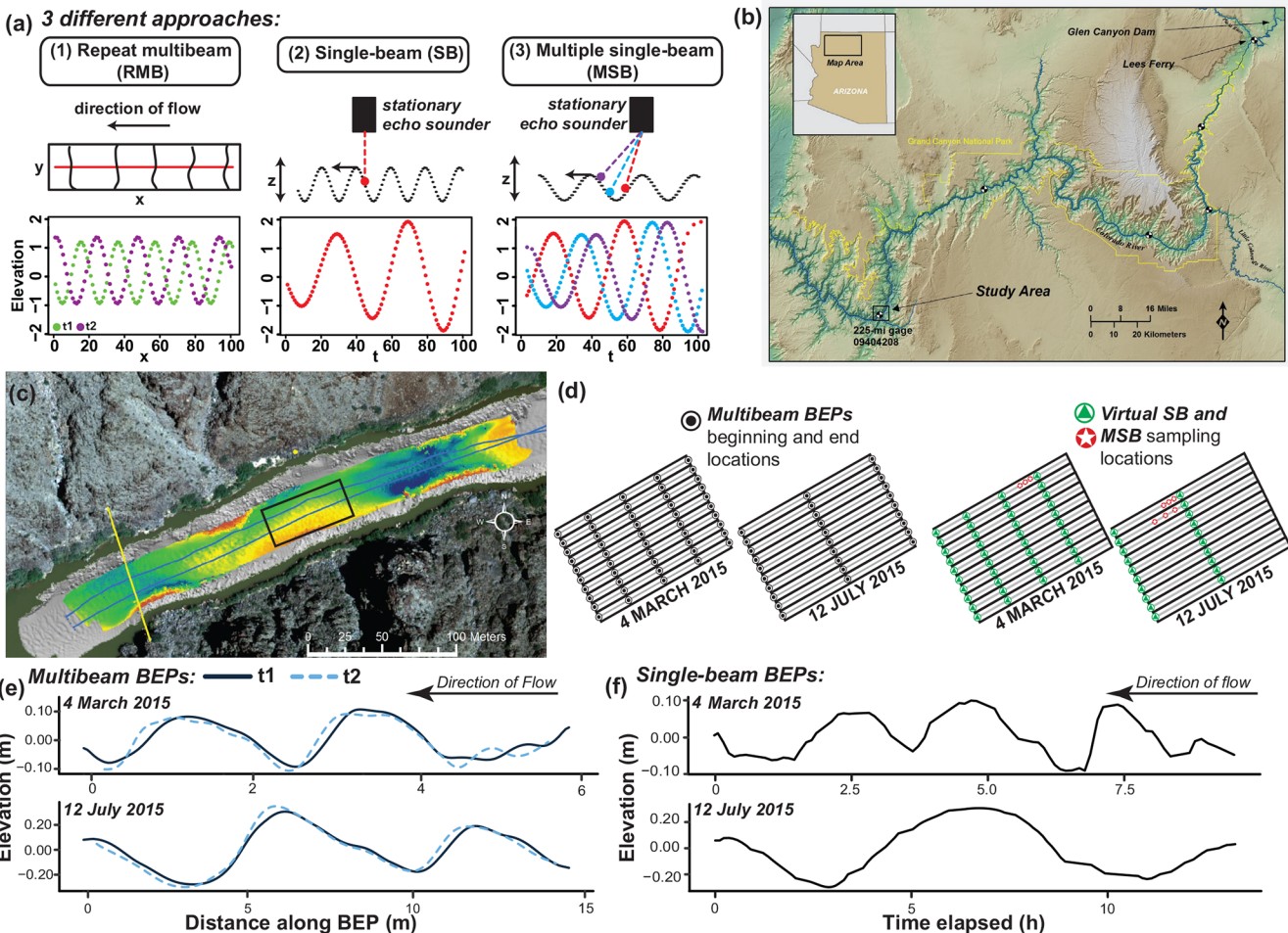

**Figure 1. (a)** Schematic of three common field methodologies for collecting BEPs (repeat multibeam, single-beam, and multiple single beam) and the types of BEPs produced by each method. **(b)** Location of study area on the Colorado River in Grand Canyon National Park. Map adapted from Kaplinski et al. (2017). **(c)** Map of study reach. Yellow line indicates the location of the Diamond Creek USGS gage. Gray area indicates area mapped with single multibeam survey. Colored area indicates area over which repeat multibeam surveys were collected (colors indicate elevations where red is high and blue is low). The blue lines that bisect the repeat multibeam survey area indicate the track lines the boat drove along in order to obtain each survey. Black rectangle indicates area in which BEPs were extracted. **(d)** BEP extraction map for multibeam, virtual single beam, and virtual multiple single beam. Examples of repeat multibeam **(e)** and single beam **(f)** BEPs for March and July datasets.

were collected using a Teledyne-Reson 7125 multibeam echo sounder, with sensor attitudes provided by a vessel-mounted inertial navigation system, and positions telemetered to the survey vessel at 20 Hz using a robotic total station situated onshore on monumented control. Data were collected with a 50 % overlap between adjacent sweeps, providing up to 1000 individual soundings per square meter. Each sounding was edited manually. Further details of this system, survey, and processing methods are given by Buscombe et al. (2014, 2017) and Kaplinski et al. (2017). The channel bed was entirely composed of fine to medium sand with no gravel patches (Buscombe et al., 2017). At each discharge, data were collected every 6–10 min for 12 h. A digital elevation model of the riverbed was produced for each survey, us-

ing coincident 0.25 m × 0.25 m TS2 grids. The March 2015 (around 283 m³ s⁻¹) and July 2015 (around 566 m³ s⁻¹) repeat multibeam surveys occurred during mostly increasing and decreasing hydrographs, respectively (Fig. 2a). The precision of the repeat surveys was high, with mean cell elevation standard deviation of 0.012 m computed over rocks known to be immobile (Kaplinski et al., 2017).

## 2.2 Extraction of bed elevation profiles

A 35 m × 30 m subsection in approximately the middle of the area surveyed by the repeat multibeam was selected for detailed bed-load analyses using repeat multibeam, single-beam, and multiple single-beam BEPs (Fig. 1c). This subsection was then divided into 40 different repeat multibeam

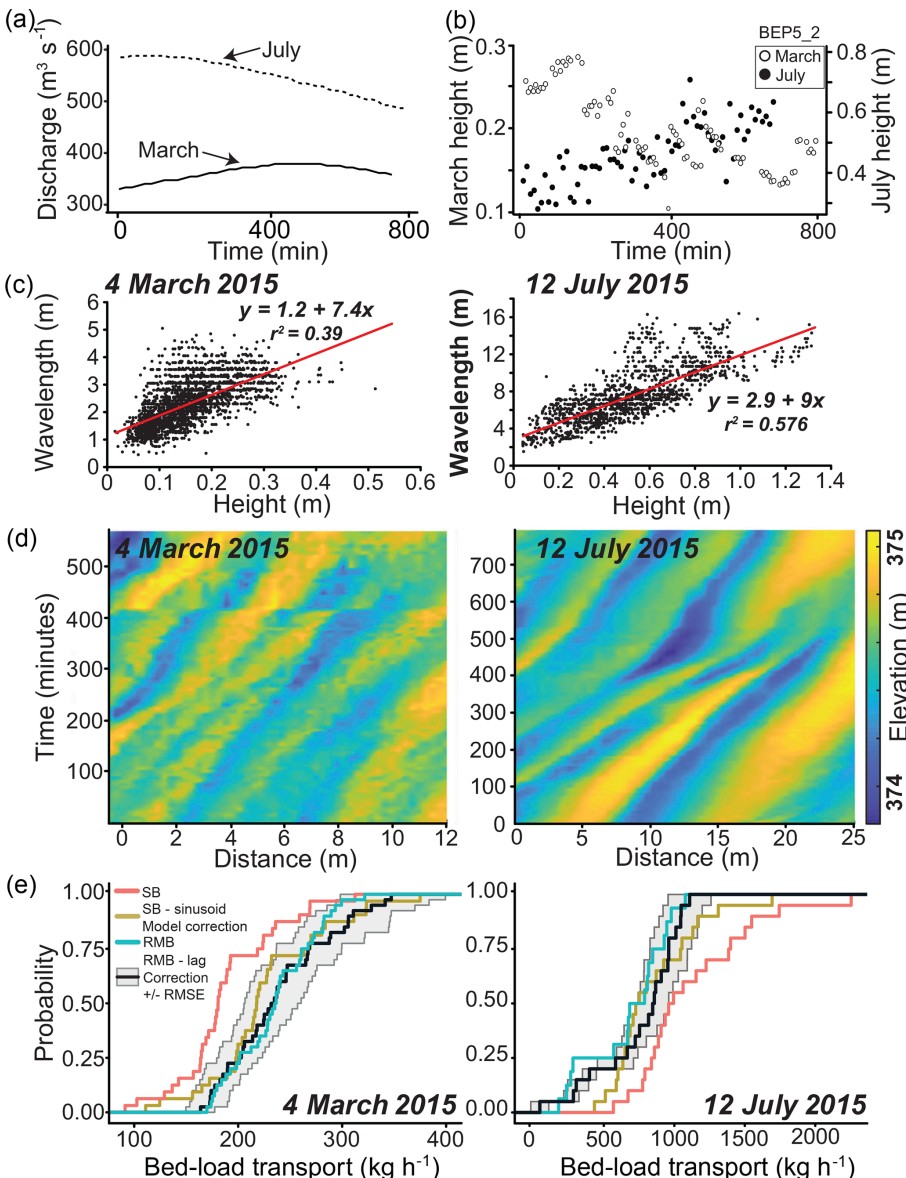

**Figure 2. (a)** Discharge during the sample time period. Dashed line is July data; solid line is March data. **(b)** Example of bedform height varying with time from a single BEP. Open circles indicate July data; closed circles indicate March data. **(c)** Wavelength versus height. Red line indicates linear regression of the data. **(d)** Space–time plots of bedform evolution. **(e)** Cumulative density plots of single-beam (SB; red line) and repeat multibeam (RMB; teal line) bed-load transport estimates with added corrections for mischaracterized lag (repeat multibeam; black line with gray area indicated root-mean-square error) and period (single beam; yellow line).

BEPs (8.67 m in length, 3.67 m lateral spacing from one another; Fig. 1d) for March and 20 different repeat multibeam BEPs for July (17.34 m in length, 3.67 m lateral spacing from one another; Fig. 1d). The length of the BEPs was determined by considering the maximum dune wavelength. All repeat multibeam BEPs were detrended using the bedform tracking tool (BTT) developed by van der Mark et al. (2008). This tool detrends each BEPs using a weighted moving average. After the BEPs are detrended, the BTT determines the zero upcrossing (i.e., points at which the profile positively crosses zero) and zero downcrossing (i.e., points at which the profile negatively crosses zero). The locations of crests and troughs are determined in the original BEP as follows: a crest is located at the maximum value between a zero upcrossing and zero downcrossing; vice versa, a trough is located at the minimum value between a zero down- and zero upcrossing. Bedform height is calculated at the vertical distance between crest and downstream trough. Bedform wavelength is calculated as the distance between two successive crests. For a

more detailed explanation of the BTT please refer to van der Mark et al. (2008) and van der Mark and Blom (2007).

Whereas repeat multibeam analyses can be carried out in two dimensions (Nittrouer et al., 2008; Abraham and Pratt, 2010; Abraham et al., 2011; Shelley et al., 2013), analyses were deliberately carried out using one-dimensional transects oriented with flow direction, so any anisotropic effects in flux (caused by dunes not aligned perpendicular to the flow) affected repeat multibeam, single-beam and multiple single-beam results equally. Virtual single-beam and multiple single-beam echo sounders were placed at the downstream end of each repeat multibeam BEP (Fig. 1d). Multiple single beams have four virtual beams, one of which is the same beam location as the single-beam virtual echo sounders. Two different beam spacings were explored: (1) 0–0.56–1.16–1.74 m and (2) 0–1.74–3.48–5.22 m.

## 2.3 Calculating bed-load transport

Bed-load transport, $q_b$ (m$^3$ TS3 s$^{-1}$), was calculated using the Simons et al. (1965) formulation based on the two-dimensional Exner equation (Paola and Voller, 2005) for bed sediment mass conservation, assuming triangular dunes:

$$q_b = (1 - p)V_c \frac{H}{2} - q_e - q_0, \tag{1}$$

where $p$ is the porosity of the sand (0.35 was used here) and $q_0$ is a constant of integration (set to zero here; see McElroy and Mohrig, 2009, for a discussion of the potential physical meaning of this term). The original formulation of Eq. (1) has been validated and extended by numerous studies (e.g., Willis and Kennedy, 1975; Engel and Lau, 1980; Havinga, 1983), most recently by Shelley et al. (2013), who proposed the addition of the $q_e$ term, defined as

$$q_e = \frac{V_c^2 \Delta t H}{2\lambda}, \tag{2}$$

where $\Delta t$ is the change in time between successive surveys. Physically, $q_e$ represents an area of underpredicted transport in the original Simons et al. (1965) formulation. Shelley et al. (2013) developed $q_e$ to account for that missing area, which is typically a small part of the flux, perhaps negligible within overall error in flux. Note that Eq. (1) is averaged over a field of dunes (i.e., the entire BEP). TS4

The primary variables in the above equations are calculated differently for each type of BEP. For repeat multibeam, $H$ and $\lambda$ are calculated directly using the BTT. $V_c$ is calculated using a cross-correlation of two consecutive BEPs (McElroy and Mohrig, 2009; Engel and Lau, 1980). Using Eq. (1), we calculated 2720 individual repeat multibeam bed-load transport estimates for March and 1740 individual bed-load transport estimates for July. These individual bed-load transport estimates were then averaged to generate 40 average daily bed-load transport estimates in March and 20 average daily discharge estimates in July.

Single-beam data consist of time-varying elevation only (Fig. 1a, f); therefore $\lambda$ must be estimated independently. This might be done by measuring dune wavelengths in the field (for example, by wading, TS5 SCUBA, or using a boat-mounted sonar or ADCP) while installing or maintaining the echo sounder. To simulate such an exercise, we use the daily average wavelength calculated by the BTT from the repeat multibeam survey directly upstream of the virtual single-beam echo sounder. Celerity ($V_c$) is

$$V_c = \frac{\lambda'}{T}, \tag{3}$$

where $T$ is the period, and $\lambda'$ is the estimated average wavelength.

For multiple single-beam data, average period and height can be measured directly from the BEPs, whereas ($V_c$) may be estimated in one of three different ways. The first, "original method", is the same as Eq. (3), in which each beam is treated as a separate BEP to produce four estimates of transport that are then averaged. The second "cross-correlation method" is to use a cross-correlation of BEPs measured by two different beams to find the spatial offset or "lag", $l$, between translated dunes:

$$V_c = \frac{D}{l \times \Delta t}, \tag{4}$$

where $D$ is the distance between sensors. In a field situation, this is constrained by practical considerations, but here we are free to vary $D$ to evaluate its effects. This method produces six estimates of bed-load transport (from six pairs of four beams), as does the third, "manual method", in which velocity is

$$V_c = \frac{D}{t_{m2} - t_{m1}}, \tag{5}$$

where $t_{m1}$ and $t_{m2}$ are manually picked times at which a crest appears at each beam.

## 2.4 Sinusoid model of bedform growth and decay

To test the effects of the adjustment of bedform height and length to changing flow conditions, a simple sinusoid model was used to simulate time-varying dune height and wavelength. Each detrended bed elevation series was approximated by

$$\eta = A \sin(B + Cx). \tag{6}$$

Dune growth/decay was controlled by varying $A$ (amplitude) and $C$ (wavelength). Dune translation was controlled by $B$ (shift). Dune wavelength was estimated from dune height according to the regressions of the relationship between bedform height and wavelength for each survey day (Fig. 2c). Using Eq. (6), sinusoid single-beam BEPs are constructed from the synthetic elevation series, $\eta$, at a single location, $x$.

These profiles were then used to calculate synthetic single-beam bed-load transport estimates using Eqs. (1) and (3). The average $\lambda$ and $H$ were used in these calculations to replicate the methods used for the virtual echo sounder experiment. Synthetic repeat multibeam bed-load transport rates are calculated using Eq. (1). We then take the ratio of synthetic multibeam bed-load transport estimates to synthetic single-beam bed-load transport estimates and use that ratio as a correction fact for our actual measurements (i.e., multiply the actual single-beam estimates by the ratio determined in the sinusoid model).

## 3  Results

### 3.1  Dune field characteristics

Bedform height averaged 0.17 and 0.36 m in March and July, respectively. Bedform wavelength averaged 2.38 and 5.1 m in March and July, respectively. Dune geometry was highly variable during both survey days, with standard deviations of bedform height (wavelength) of 0.05 m (0.4 m) in March and 0.2 m (2.7 m) in July. Discharge along the Colorado River in the Grand Canyon fluctuates daily as a result of daily release flows from Glen Canyon Dam. In response to changes in discharge, bedform size almost doubled over the course of the survey in March and almost halved over the course of the survey in July (Fig. 2b). Due to the greater discharge, the bedforms in July are larger in height and wavelength (Fig. 2c) compared to those in March. A space–time plot of bed elevations shows bedform heights and wavelengths increasing over the duration of sampling in March (Fig. 2d). Bedform crest traces become less frequent as bedform troughs deepen. In contrast, a space–time plot of bed elevations in July shows bedform heights and wavelengths decreasing throughout the survey period (Fig. 2d).

### 3.2  Repeat multibeam vs. single beam

We consider the repeat multibeam-derived bed-load estimates to be the most accurate because the superior spatiotemporal coverage of these data allows for simultaneous resolution of $V_c$, $H$, and $\lambda$. Single-beam-derived daily bed-load transport rates are underestimated relative to repeat multibeam in March and overestimated in July (Fig. 2e). This could be caused either by mischaracterization of $V_c$, $H$, or $\lambda$ in either repeat multibeam or single-beam calculations, or in both.

The most likely source of error in the repeat multibeam calculations occurs when calculating $V_c$. To investigate whether cross-correlation correctly measured translation of dunes, $l$ was manually calculated from repeat multibeam BEPs by picking the locations of crests and tracking them and then used to calculate bedform celerity. This showed that cross-correlation-derived $V_c$ values were underestimated in both March and July (Fig. 3a). This underestimation is much larger in July, when dunes were adjusting to decreasing flow conditions and deforming at a greater rate. This result indicates that caution should be exercised when using cross-correlation to derive $V_c$, especially during higher transport stages. The regressions between manual and cross-correlation computed $V_c$ (Fig. 3a) are used to calculate a lag-corrected celerity and lag-corrected bed-load transport rates for the repeat multibeam data (Fig. 2e). Correcting repeat multibeam estimates for cross-correlation lag errors results in 1.6 % and 33.9 % error for March and July, respectively. Percent error will be expressed relative to repeat multibeam-corrected lag flux estimates for the remainder of this paper.

Even with the lag-correction applied, discrepancies exist between repeat multibeam and single-beam bed-load transport estimates due to errors estimating $V_c$ from estimated wavelength and observed period. In March, period computed from single-beam data is overestimated relative to repeat multibeam period, causing $V_c$, and therefore transport, to be underestimated. The opposite is true for the July data (Fig. 3b). These discrepancies in observed period are likely linked to the bed responding rapidly to unsteady flows during each survey (Fig. 2a), with changes in discharge causing commensurate changes in $H$ (Fig. 2b) and $\lambda$ (Fig. 2c). This suggests that dunes adjusting to unsteady flow conditions apparently distort the period observed in the single-beam data, which would invalidate the assumption made in Eq. (3) that the daily average wavelength (or any invariant measure of wavelength) is representative.

### 3.3  Single-beam correction from sinusoid model

To test the above hypothesis, synthetic repeat multibeam and single-beam bed-load transport estimates calculated from bedform growth/decay models are compared. Single-beam BEPs of growing and decaying sinusoids display significantly different distributions of period compared to the assumption of constant bedform wavelength (Fig. 3c). When bedforms are adjusting to changes in flow conditions the period recorded by the single-beam profile changes to either be longer or shorter depending on whether the dunes are growing or decaying, which affects bed-load transport measurements. As dunes grow or decay, the ratio of synthetic repeat multibeam to synthetic single-beam bed-load transport increases or decreases, respectively. In the sinusoid model, the maximum ratio of synthetic repeat multibeam to synthetic single-beam bed-load transport rates for growing and decaying dunes is 1.2 and 0.75, respectively. Applying these ratios as correction factors to the single-beam estimates (i.e., multiplying the single-beam estimates by these ratios) generates sine-corrected single-beam transport estimates (Fig. 2e), resulting in a decrease of the discrepancy between repeat multibeam- and single-beam-derived bed-load transport rates from 45.3 % to 27.7 % in March and from 38.9 % to 10.7 % in July.

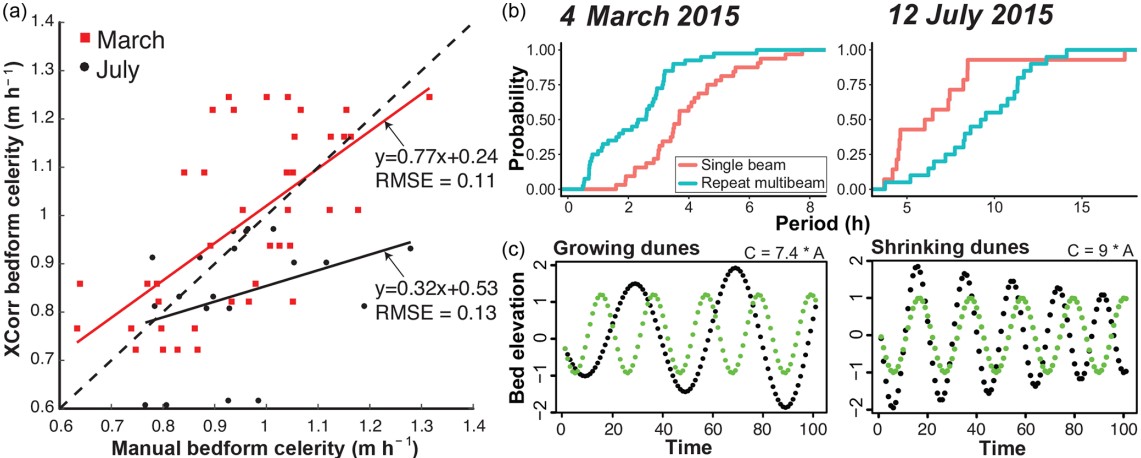

**Figure 3. (a)** Bedform celerity calculated using a manually picked lag versus a lag produced using a cross-correlation algorithm. The best fit linear regression for each survey day serves as a correction factor for repeat multibeam celerity estimates. Dashed line represents 1 : 1 relation. RMSE signifies the root-mean-square error. **(b)** Cumulative density plots of period measured from single-beam and repeat multibeam BEPs. **(c)** Sinusoid model showing what single-beam BEPs would look like if dunes growed/decayed (black) or if dunes remained the same size through time (green).

## 3.4 Repeat multibeam vs. multiple single beams

Another potential practical solution to minimizing the distortion of period in single-beam surveys is to use multiple single-beam echo sounders in a spatial array (Fig. 1). By increasing the spatial resolution of bed elevation data, multiple estimates of bed load may be obtained, as well as multiple options for computing $V_c$ (Eqs. 3 through 5), two of which (Eqs. 4 and 5) do not require a priori estimation of bedform wavelength ($\lambda'$). We expect the period recorded by each beam to be similarly affected by growing/decaying dunes as were the single-beam periods. We therefore apply the same sine correction from above to multiple single-beam flux estimates calculated with the "original method". Figure 4a shows these results for the beam spacing of 0, 0.56, 1.16, and 1.74 m for the three methods for computing celerity, and Fig. 4b shows the bed-load transport estimates using a larger beam spacing for the July data only. The original method of calculating celerity (Eq. 3) produces an average percent error of 13.3 % and 15.8 % in March and July, respectively, suggesting that increasing the number of beams and incorporating a sinusoid correction can mitigate discrepancies with repeat multibeam estimates.

The cross-correlation method (Eq. 4) systematically overestimates bed-load transport in both March (43.4 % error) and July (108.4 % error), suggesting that the lag is systematically underestimated, and hence overestimating celerity. The manual method (Eq. 5) yields a small mismatch between multiple-single-beam-derived `CE1` and repeat multibeam-derived bed load in March (1.3 % error) but a 62.9 % error in July. This could be related to beam spacing, because the bedforms (and bed-load mismatches) in the July data are much larger than those in March. This could

cause greater celerity because only between 10 % and 30 % of the dune wavelength is being captured by the multiple single beam with the smaller sonar spacing, increasing to 30 %–100 % with the larger spacing of 0–1.74–3.48–5.22 m (Fig. 4b). However, increasing beam spacing does not fully resolve discrepancies between repeat multibeam and multiple single-beam bed-load estimates (e.g., the manual methods has a 36.6 % error and the cross-correlation method still systematically overestimates bed-load transport; Fig. 4b). This suggests another factor is contributing to the observed discrepancies, most likely temporal resolution.

Using a linear interpolation we increase the temporal resolution of the data from 6 to 3 min. At this new sampling frequency, the original method yields a 2 % error in March but continues to overestimate bed-load transport in July (67.5 % error; Fig. 5). Increasing the temporal resolution of the data results in more accurate estimates of lag. The cross-correlation method yields a 6.8 % error in March and a 16.3 % error in July (Fig. 5), suggesting that the temporal resolution of the multiple single-beam data will cause variation in cross-correlation-derived estimates of $V_c$.

## 3.5 Lateral variations in calculated bed-load transport rates

We also consider how bed-load transport rates vary with distance across the channel (Fig. 6). During the March survey, mean bed-load transport rates were slightly higher on river left compared to river right (Fig. 6a). We see a similar pattern in mean bedform height (Fig. 6b) but no recognizable pattern in mean bedform celerity (Fig. 6c). In the July survey, mean bedform transport rates are slightly larger on the on river right than on river left (Fig. 6a). This is again re-

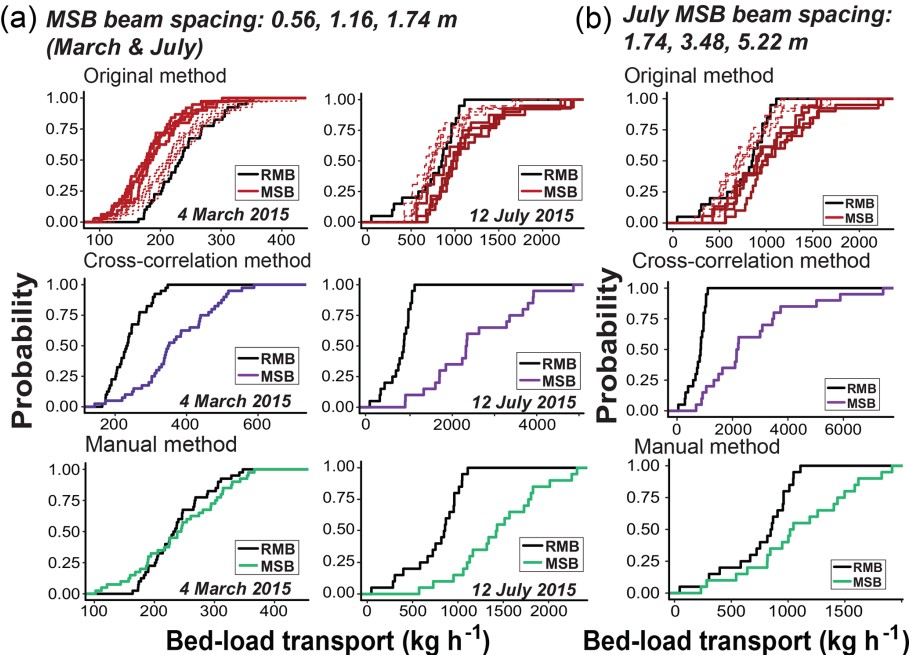

**Figure 4.** CDFs of lag-corrected repeat multibeam (RMB; black) and multiple single-beam (MSB) bed-load transport estimates using the original (red), cross-correlation (purple), and manual methods (green) to calculate bedform celerity for the multiple single-beam profiles. Dashed red lines are sine-corrected estimates. **(a)** Multiple single-beam beam spacing of 0–0.56–1.16–1.74 m for both March and July. **(b)** July bed-load transport CDFs using a larger beam spacing of 0–1.74–3.48–5.22 m.

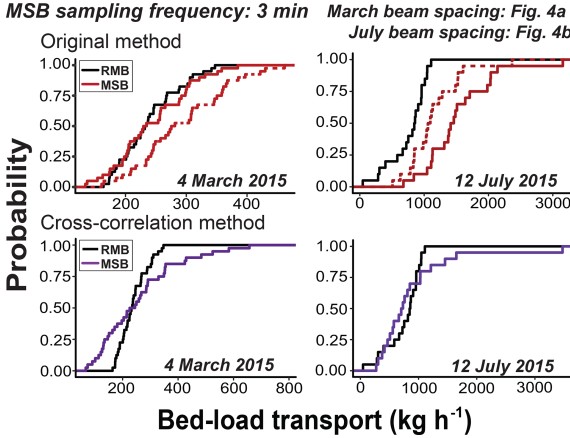

**Figure 5.** CDFs of lag-corrected repeat multibeam (RMB; black) and multiple single-beam (MSB) bed-load transport estimates using the original (red) and cross-correlation (purple) to calculate bedform celerity for the multiple single-beam profiles. Dashed red lines are sine-corrected estimates. Sampling frequency is 3 min. March beam spacing is the same as Fig. 4a. July beam spacing is the same as Fig. 4b.

flected in the bedform height distributions (Fig. 6b) but not the bedform celerity distributions (Fig. 6c). We also observe changes in the shape of the distributions of bedform transport rates and bedform heights with distance across the channel. Near the banks of the channel, our bedform transport rate

and bedform height distributions have a much larger range whereas these distributions have a much tighter range in the center of the channel. This is most likely caused by varying amounts of superimposed bedforms across the channel and suggests that the bedform decay process can be heterogeneous in the cross-channel direction.

The ratio of single-beam to repeat multibeam bed-load transport rates also changes with lateral position along the channel (Fig. 7). For this analysis we compare daily mean lag-corrected repeat multibeam bed-load transport rates and daily mean single-beam bed-load transport rates. In March, we find the highest agreement between the two methods in the center of the channel, with high discrepancies between the two methods near both banks. In July, the highest agreement between the two methods is also in the center of the channel although only proximity to one bank (river left) showed significant discrepancies in bed-load estimates.

## 4 Discussion

BEPs recorded by repeat multi-, single-, and multiple single-beam sonar methodologies produce different estimates of bed-load transport, but practical steps can be taken to reduce the mismatch.

Significant errors in bed-load transport rates can arise for two main reasons: (1) cross-correlation-derived repeat multibeam bedform celerity estimates can show systematic bias,

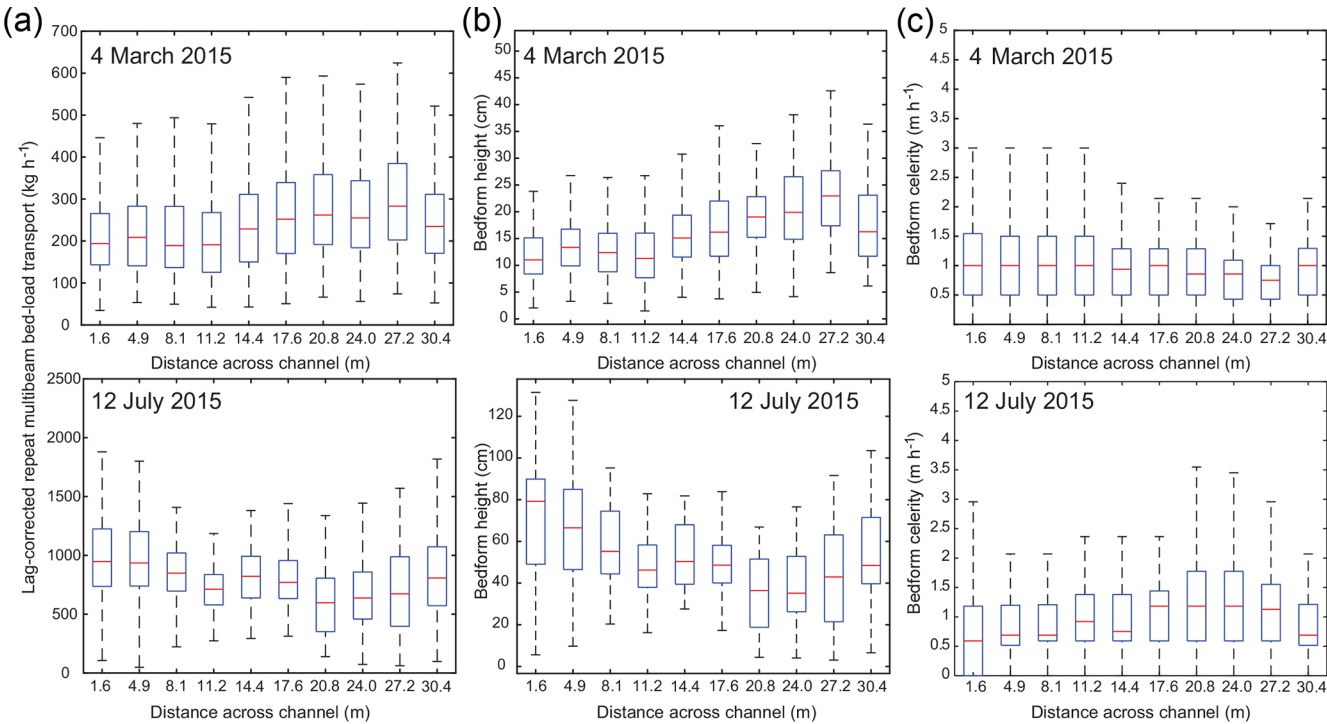

**Figure 6.** Boxplots of lag-corrected repeat multibeam bed-load transport estimates **(a)**, bedform heights **(b)**, and bedform celerities**(c)** with varying distance across the channel (from river right to river left).

and (2) dunes can grow/shrink in response to unsteady flow conditions or varying sediment supply (Martin and Jerolmack, 2013; Rodrigues et al., 2015). Caution should be exercised when using cross-correlation to derive dune celerity, especially during higher transport stages and for relatively large time increments between successive measurements. It is good practice to check lags estimated using cross-correlation with manual measurements in order to compile a relationship that can be used to correct for systematic bias in estimated lag (Fig. 3a).

Using single-beam BEPs, as dunes grow, transport is underestimated because period is overestimated. As dunes decay, transport is overestimated because period is underestimated (Fig. 3b). It is therefore important to understand the timescales over which dunes size is responding to flow in order to assess the relative effect period distortion may be having on the bed-load estimates. A sinusoidal growth model is proposed that accounts for geometric effects on bed-load transport in unsteady flows, using measured dune heights and translations and a scaling relationship to predict dune wavelength from dune height (Fig. 3c). Such a scaling relationship could be compiled over time for a specific single-beam deployment and applied retroactively to entire time series of BEPs. The sinusoid model could be applied in any operational setting where temporal variations in dune wavelength and a dune height–wavelength scaling relationship exist.

A less generally applicable extension to this procedure TS6 could involve modeling the spatiotemporal evolution of the bed more explicitly using Fourier series (e.g., Guala et al., 2014). Guala et al. (2014) demonstrate a frequency dispersion in the relationship between dune celerity, $V_c$, and wavelength, $\lambda$, because small dunes tend to move faster than larger ones. This does not bias our computed bed-load fluxes from multibeam data since we use time series of bedform statistics from $\eta(x, y, t)$; however, it does place limits on any calculation of equivalent statistics from $\eta(t)$ because it requires assuming a model that relates average $V_c$ with average $\lambda$, or rather that the functional form between them does not vary in time, which may not be strictly true. TS7

In this study, accounting for temporal changes in dune geometry accounted for $28.9\,\mathrm{t\,d^{-1}}$ (March) and $134.8\,\mathrm{t\,d^{-1}}$ (July) in daily bed-load rates computed using single beam, or $17.6\,\%$ (March) and $28.3\,\%$ (July) compared to lag-corrected repeat multibeam-derived rates. It is worth noting that real single-beam echo sounders can operate at a finer temporal scale than we are able to approximate in our virtual experiment. Our analysis indicates that single-beam measurements made at a fine enough scale could greatly reduce the error in bed-load flux estimates.

Increasing the spatial resolution of the bed elevation data by using a multiple single-beam system does not necessarily improve upon single-beam transport estimates. Multiple single-beam transport estimates do not suffer from distor-

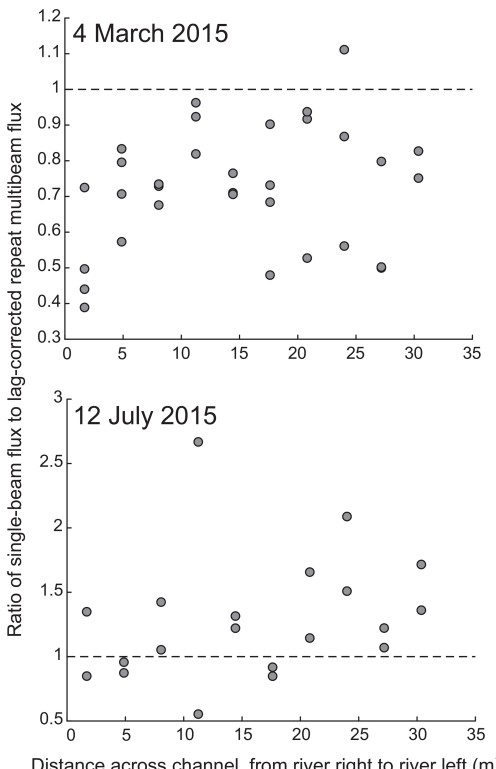

**Figure 7.** The ratio of daily mean single-beam bed-load transport estimates to daily mean lag-corrected repeat multibeam bed-load flux estimates with varying distance across the channel (from river right to river left). Dashed line indicates a ratio of 1, where we expect comparable magnitudes of bed-load transport estimates between the two methods.

tions in period caused by changing dune wavelength but are sensitive to both beam spacing and sample frequency (Fig. 4). Ideally, sonar beams should be spaced such that a large proportion of the dune wavelength is sampled (Fig. 4b), although this is not always practical, especially in shallow water. If dune wavelengths change significantly according to flow, designing sampling to be optimal for a particular wavelength would not be recommended. A more effective approach to maximizing multiple single-beam-derived bed-load accuracy is to adjust sampling rate (Fig. 5), calibrated in relation to a known range of dune migration rates. This is especially helpful for dune celerity estimates based on cross-correlation (Fig. 4a). We found the most accurate way to measure dune celerity from multiple single-beam data is to measure time elapsed between successive dune crests.

Our analyses also suggest that lateral position along the channel can significantly affect single-beam bed-load transport estimates (Figs. 6, 7). When mounting single-beam or multiple single-beam systems, its important to be cognizant of proximity to channel margins. We found that the center of the channel yielded the most comparable magnitudes of bed-load transport rates between sampling methods. In our

study area, the center of the channel is very near the thalweg of the channel. In rivers in which the thalweg differs significantly from the center of the channel, we suggest mounting the echo sounder in a position near the thalweg if possible.

## 5   Conclusions

All practical methods for estimating bed load from dune observations suggested to date require a time series of bed elevation changes associated with dune migration, either $\eta(t)$ using a single-beam echo sounder or $\eta(x, y, t)$ using a multibeam sonar. In lieu of a parameterization for bedform migration speed appropriate for field scales, forced by routinely measurable quantities such as discharge, ambiguity between wavelength and period can negatively affect bed-load transport estimates from single-beam echo sounders. Here, a simple model to simulate changes in dune wavelength was sufficient to resolve much of the ambiguity. Repeat multibeam-derived elevation time series are a more accurate means with which to estimate bed load than using a single beam or multiple single beams, because the superior spatiotemporal coverage of these data allows for simultaneous resolution of $V_c$, $H$, and $\lambda$. However, there are significant practical advantages to using single-beam or multiple-single-beam systems over repeat multibeam, and their capacity to monitor bed load over long periods may in some situations outweigh any disadvantages to do with greater errors in instantaneous bed-load flux. We have offered a case study and practical guidelines to maximizing the efficacy of comparing bed-load transport estimates derived from different sampling methodologies, which collectively will guide design of optimal bed sampling strategies for tracking dunes in rivers.

**Code and data availability.**  Data and related codes are available at https://doi.org/10.5967/M02J6904 (Leary, 2019).

**Author contributions.**  KCPL designed the virtual experiments, developed code, and performed the simulations. DB provided invaluable insight and helped with data analysis. KCPL prepared the manuscript with contributions from DB.

**Competing interests.**  The authors declare that they have no conflict of interest.

**Disclaimer.**  Any use of trade, product, or firm names is for descriptive purposes only and does not imply endorsement by the United States government. TS8

**Acknowledgements.**  We would like to thank Robert Mahon, Kory Konsoer, one anonymous reviewer, and Associate Editor Claire Masteller for their thoughtful feedback and constructive

reviews. Thanks are owed to Matt "The Colonel" Kaplinski for leading the field surveys and processing all multibeam data, and to Bob Tusso, Erich Mueller, and Tom Ashley for field support. Dave Topping, Brandon McElroy, Paul Grams, Dave Dean, Mark Schmeeckle, and Kelin Whipple provided useful discussion.

**Financial support.** Data collection was funded by the Glen Canyon Dam Adaptive Management Program administered by the United States Bureau of Reclamation, through a cooperative agreement with the United States Geological Survey Grand Canyon Monitoring and Research Center (grant nos. G19-000704 and G18-AC00038).

**Review statement.** This paper was edited by Claire Masteller and reviewed by Robert Mahon, Kory Konsoer, and one anonymous referee.

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

**Remarks from the language copy-editor**

**Remarks from the typesetter**