# Peer review of "Estimating Sand Bedload in Rivers by Tracking Dunes: a comparison of methods based on bed elevation time-series"

_Earth Surface Dynamics, 2019_

## Referee Comment (RC2) · Robert Mahon (Referee) · 23 Sep 2019

The authors present a systematic comparison of bedform bedload measurement techniques using a unique dataset. Using field data, as opposed to flume data as is often the case, the authors are able to investigate some of the complexities associated with systems evolving under unsteady flow conditions. The ultimate outcome of this paper can inform decisions on both multibeam sampling and processing strategies as well as the placement of singlebeam echosounder instrumentation on rivers to monitor bedload flux. Thus the results of this paper are broadly relevant to river managers as well as to academic geomorphologists.

[Figure]

The overall flow and structure of the manuscript are quite clear. Figures are well placed into the manuscript context and are appropriate for fully describing the nature of the work. While I have no concerns that fundamentally call into question the nature of the science being done, there are a number of points which the authors could clarify or analyses that could be bolstered by more complete discussion. These comments are below:

I would like to see a description of the methods used to extract height and wavelength data from the BTT toolbox as it is a fundamental operation to the analysis in the paper. There are several methods for calculating these parameters, each of which have their respective advantages and disadvantages so it would be good for the authors to describe why the calculations employed in this toolkit are appropriate to their system.

Were bed elevation surveys corrected for apparent dilation as a function of the time between start and end of each multibeam survey? If not, was this considered and determined to be a negligible effect? See McElroy dissertation 2009, p. 44 (URI: http://hdl.handle.net/2152/15117).

A figure demonstrating the cross-correlation results would be good to show, as a lot of discussion is based on issues resulting from velocity calculations.

In Page 5 Line 6 the method for estimating wavelengths for the singlebeam experiment is described as the daily average from the repeat multibeam. I wonder if this introduces potential for extra accuracy for this method that may not be possible in a situation in which a single beam fixed echosounder would be employed. I would suggest more discussion of when a situation would arise where you have a measurement or a daily average of bedform wavelengths but only a single beam profile to estimate flux from. An alternative formulation might be to estimate wavelength using a height-wavelength relationship such as Bradley and Venditti, 2017 as this might be a more realistic representation of a likely application (i.e. a deployed single beam sensor established for continuous monitoring).
What did the manual process entail for determining bedform velocity? Were you picking crests and tracking them? Looking at the slopes of the forms in the $\eta$(x,t) field (e.g. in Figure 2D)? It would be critical to determine whether the manual method itself includes any potential sources of bias in order to interpret its relation to the cross-correlation results.

I wonder if other methods for calculating bed velocity might be more appropriate than the cross-correlation method for this application, particularly given the unsteady flow conditions investigated. One example from Ganti et al., 2013 (doi:10.1002/jgrf.20094), their eq. 5 to compute the local velocity based on dividing the temporal change in local elevation by local slope at all points on the bed.

Were any physical bedload samples collected during the multibeam campaigns to compare with the ranges of flux measurements?

Some discussion is warranted of whether the bedform bedload equation of Simons et al., is even geometrically appropriate in situations where bedform growth/decay is occurring. I don't believe they considered this in their original work, and I am not aware of any later publications that show the validity of this method for non-steady bedform fields.

Along similar lines I would encourage the authors to consider incorporating, or at least explaining the inappropriateness for their application, the insights from Guala et al. (2014, their Section 4 paragraph 2 in particular; doi: 10.1002/2013JF002759) in joint averaging of the elevation and velocity values.

While somewhat outside the scope of the review of the paper itself, I should note that the license type given to the dataset and code hosted in the SEAD repository is potentially quite restrictive to some river management uses and researchers, given that it does not allow commercial use or any derivatives. This may be less important for the data itself, but it may heavily limit the use of this work to have code that cannot be modified. A share alike restriction, for example, would make this more accessible.

Line Comments: The following line specific comments are non-critical to the science of the manuscript and are meant to help improve readability or clarity. Page 1 Line 2: "remains elusive" is relatively non-concrete and feels dismissive of the wealth of literature and practice on field-scale bedload measurement techniques spanning half a century or more. Page 1 Line 14: references are missing at "(e.g. ?)" Page 1 Line 20: References such as Simons et al., 1965 and others don't explicitly derive from the exner equation, per se. They are derivations of mass conservation but not necessarily predicated on Exner's formulations.

Page 2 Line 1: Simons wasn't the first to show this, as written. For example, Bagnold 1941, Chapter 13 derives a similar formulation, albeit with some geometric inaccuracies. I suggest simply removing the word "first" from the sentence. Page 2 Line 9: remove comma after "...discharge conditions," Page 2 Line 12: is there a reference for "...bedload flux estimated from translating dunes remains one of the most accurate..."?

Page 4 Line 14: ISDOTTv2 is not a familiar/common tool since it is not public. If you wish to include this statement, it would be good to describe what that tool is and why it would be useful here. Otherwise I would suggest removing it. Page 4 Line 16: please describe the "missing triangles" correction. Page 4 Line 22: consider rewording as it states a 1965 reference is based on a 2005 reference.

Page 7 Line 14: "...for growing (shrinking) dunes is 1.2 (0.75)." I suggest rewording to "...for growing and shrinking dunes is 1.2 and 0.75, respectively."

Figures: for figures 1, 2 and 4 there are abbreviations used which would be helpful to have defined in figure captions so the reader doesn't have to remember or find from the text. BEP, RMB, SB and MSB are all used. Additionally, Xcorr and RMSE are used but not defined in captions or in the text body.

---

## Referee Comment (RC3) · Kory Konsoer (Referee) · 1 Oct 2019

In "Estimating Sand Bedload in Rivers by Tracking Dunes: a comparison of methods based on bed elevation time-series", the authors present a systematic comparison for different approaches for estimating bedload transport based on dune migration. The methods compared rely on repeat multibeam echo sounding surveys from a reach of the Colorado River during two different field campaigns that exhibit different discharges. The multibeam surveys provide the base data, and three different subsets from the data are selected. The three datasets used in the comparison are, 1) longitudinal transects of bed elevation from the full multibeam surveys, which provide spatial data series, 2)

[Figure]

extraction of bed elevation at a single point over time (temporal), and 3) extraction of bed elevation at multiple points over time (temporal). The authors also include synthetic sinusoidal signals that are used to evaluate bedform dynamic of growing/shrinking size that would occur during unsteady flows.

Overall the paper is well written and organized, and the presentation of the results is very clear. The topic of this paper is also of great importance as river scientists still struggle with determining best practices for quantifying bedload transport rates. However, I would recommend addressing a few issues related to the methods and discussion before the manuscript should be accepted for final publication. I outline these below.

Although the data are measured using a multibeam echo sounder, the dataset is not fully utilized and instead only bed elevation profiles are extracted. Thus, the comparisons are essentially spatial series of single beam, stationary single beam, and stationary multi-single beam. It is stated that the reason for this is to account for anisotropy among the different methods equally (page 4, lines 11-14), which is understandable. However, as is stated more than twice throughout the manuscript, multibeam surveys are considered the most accurate due to the high spatio-temporal resolution, yet are not being used to their full potential. Why have you decided not to include the full three-dimensionality of the multibeam survey when considering sediment transport? If you consider this to be most accurate, then you could conceivably have a fourth method using the repeat multibeam surveys as two dimensional differencing compared to the three "single beam" methods presented in the paper.

Similarly, it appears as though all the repeat multibeam bed elevation profiles have been averaged into a single value for the area of interest. Why not keep these separate and evaluate the comparisons spatially? From the bed elevation raster shown in figure 1 there appears to be quite a difference in elevation and bedform size from the left bank (higher bed elevation) to right bank (lower bed elevation). Is there a systematic difference in the comparisons from left to right? If so, is it related to bedform

dimensions?

This spatial information would be extremely relevant for the discussion section. In particular, one of the topics I felt was missing from the discussion was how the findings of this study can be used to provide insight on where stationary single beam sensors could be installed. My understanding is that most single beam sonars are attached to bridge piers or off banks/docks. If a spatial component of comparison is included in this paper, it would be possible to inform deployments in future studies. Do your comparisons show less agreement between the methods closer to the bank? These are questions easily answered from your dataset without much additional analyses.

Could you provide more information on how the cumulative density plots are prepared? It is stated on page 4 line 30 that Eq. 1 is averaged over a dune field. There is no mention of how the CDF are prepared. How many bed elevation profiles are needed before a 'stationary' average bedload transport rate is obtained? How far apart do the lines need to be? Answers to these questions could help guide surveys using boat-mounted single beam sonars. (it is stated that this is not of concern for the paper, however the extracted profiles from the multibeam survey is essential that).

There is reference to a figure 5 and figure 6 on page 6, but figures are only 1-4. I have attached an annotated pdf with other technical issues. Please see for grammar and other comments.

Please also note the supplement to this comment:
https://www.earth-surf-dynam-discuss.net/esurf-2019-38/esurf-2019-38-RC3-supplement.pdf

---

## Author Comment (AC1) · 8 Nov 2019

*We would first like to thank the anonymous reviewer for their thoughtful review. In the below document, the reviewers comments are in black; our responses to reviews are in blue italics.*

This manuscript compares three different techniques to track bedforms and estimate bedload transport rates. This paper could be very useful for scientists who consider estimating bedload transport rates by bedform tracking, even though the paper does not include new methods. In general, the introduction, discussion and conclusion are very clear and informative. However, the methods and results are sometimes more difficult to read and need extra sentences to explain the concepts and how the conclusions are derived from the results. See my comments below.

Specific comments:
- P2, L30: "(also called altimeters...)", depending on the importance of this message, should this be mentioned earlier in the text and not between brackets?
*We have removed this information from the text, because we feel "stationary single beam echosounder" is well understood.*

- What is the difference between the second and third research question at the end of the introduction? It reads like it is the same question, but then the other way around.
*The second question asks if changes in bedform size and shape affect measurements from different sampling methods. The third question asks how it affects the measurements. We have reworded these in the text for clarity.*

- What is the possible influence of the study area on the results? In the introduction there is a distinction between shallow and deep rivers when mentioning the practical use of the multibeam and single beam, is the study area shallow or deep?
*The study area is quite deep (6-9 m depending on discharge) so multibeam is a practical choice. This information has been added to the revised manuscript.*

In the results, it is mentioned that there is a daily discharge variation that influences the bedform dimensions, how extreme are these discharge variations compared to other rivers and would this influence the advice in the discussion?
*On a daily timescale, these changes are pretty significant compared to other rivers. That being said, sediment and water discharge conditions vary continuously in natural rivers causing bed morphology to often be out of equilibrium with prevailing flow conditions. As we state in our introduction, numerous field studies suggest that bedform disequilibrium is likely the norm rather than an exception in natural river systems (e.g., Frings and Kleinhans, 2008; Julien et al., 2002; Ten Brinke et al., 2009; Wilbers and Ten Brinke, 2003). It is for this reason that the sinusoid model was developed, to account for non-stationarity in the flow causing increases or decreases in bedform dimensions in time. The sinusoid model is suggested as a utility for such situations to minimize the error in single beam style estimates of bedload flux from stationary echosounders in unsteady flow.*

- Is there an effect expected of using virtual single and multiple single beam profiles based on the multibeam data, instead of measuring it separately and thus independently in the field?
*There are arguments both ways: (1) The benefit of this virtual experiment is that we know the virtual single beam echosounders are measuring the exact same bedforms the multibeam is measuring, so independent measurements might have more error. (2) That being said, real single beam echosounders can operate at a finer temporal scale than we are able to approximate in our virtual experiment. As shown by our virtual experiment, temporal resolution makes a big difference in bedload flux estimates. If independent single beam measurements were made a fine enough scale, this could greatly reduce the error. We also were limited by our field site, which has no bridge access. So at this location we were not able to take independent single beam measurements. We have added text to the discussion in regards to this question.*

- Section 2.2, L11-14: fluxes caused by dunes that are not aligned perpendicular to the flow are ignored to be able to compare the results between multibeam and single beam. How much is this expected to influence the estimated bedload transport? Is this taken into account in other multibeam studies? The effect of varying dune dimensions due to disequilibrium with the flow is taken into account, should transport direction be taken into account as well?
*Single beam echosounders would not be able to assess transport in other directions besides streamwise. However, the multibeam data are chosen specifically to be comparable to single beam data. As such, it isn't in the scope of any paper to look for directionality in a single beam trace.*

- Section 2.2, L14: "we have chosen not to incorporate the ISDOTTv2": add a short explanation of what this method entails.
*We have removed this section of the text on the advice of another reviewer.*

- Section 2.2: I think the readability of this section could be improved by removing some of the information between brackets and incorporate it in the sentence. E.g. line 9-10, line 13. This might be a personal preference, but in general it feels like there is important information between brackets throughout the paper and therefore this information seems less important and less clear. Another example is the definition/cause of bedform equilibrium in the first sentence of section 2.4. I think some definitions and explanations will be clearer when this is explained in extra sentences.
*We have reworded and reorganized this section for clarity.*

- Section 2.3, L28: what is the physical meaning of $q_e$ and why does it need to be added? Could you add a short explanation?
*$Q_e$ represents an area of underpredicted transport discussed by Shelley et al. (2013). The area represented by C in Shelley et al. (2013) figure 1 (see below) is not accounted for using the original method of Simons et al. (1965). $q_e$ is the area of triangle D and therefore accounts for that missing portion. We have added a short explanation of $q_e$ to the text. It is a very small contribution to total computed flux.*

[Figure]

**Fig. 1.** Dimensions of a triangular sand dune

- Section 2.3: Is it possible to calculate an estimated average wavelength from the time series since you can estimate celerity from this? Would it differ a lot from the spatial estimate?
*It's not possible to calculate an average wavelength from the single-beam timeseries because it only measures elevation through time. Guala et al (2014) showed that bedform space-time substitution in this way cannot work; imposing a relationship between the wavenumber and frequency spectra breaks down because small bedforms travel faster on average than large bedforms.*

- Section 2.4: this section misses an explanation of why the bedform disequilibrium is determined Even though this is mentioned before, it would help the reader to repeat this here shortly. Furthermore, it is explained how equation 6 is used to calculate synthetic bedload transport estimates, but not how this is used to determine bedform disequilibrium.
*We are empirically accounting for bedform adjustment to changing in flow (i.e. bedform growth and bedform decay). We have updated the text to reflect this.*

- Where are figures 5 and 6?
*Those figure references were for a previous version and were mistakenly left in this version. We apologize for the confusion and have corrected the manuscript to reflect the correct figure references.*

- Section 3.2, L 24-26: how are the lag-corrected bedload transport and celerity calculated? And the errors? This might be visible in figure 5 and 6, but the pdf only shows figures 1 to 4.
*Please see figure 3A for the regressions and r-squared values mentioned in page 6 lines 22-23.*

- Section 3.3: I don't really understand yet how the sinusoid model is used to correct the data. I think this would be clearer if the method section 2.4 explains this better. What do you mean with the ratio between synthetic multibeam and synthetic singlebeam?
*We take the ratio of synthetic multibeam bedload transport estimates to synthetic single beam bedload transport estimates and use that ratio as a correction fact for our actual measurements (i.e. multiply the actual single beam estimates by the ratio determined in the sinusoid model). We've added and reworded the text to make this more clear.*

- Section 3.3, L 16: is this compared to the multibeam that is corrected for cross- correlation lag errors?
*This is in the sinusoid model. We have reworded for clarity.*

- Figure 4B: There is only one line for the multiple single beam? Shouldn't there be more lines for different spacings?

*We only use a different spacing for the July data. You can compare the CDFs to Figure 4A for the smaller beam spacing.*

Technical corrections
*We have corrected the below technical corrections in the main text.*

- P1, L14: There is a "?" instead of a source
- Figure1C: I do not see the grey section that indicates the area that is mapped with the single multibeam survey.
- Section 2.2, L8: Did you define BEP before this? You can for example add "(BEP)" at line 2 of this section
- Figure 2: there seems to be a caption missing to panel D.
- Figure2B: what is BEP5_2?
- Figure2C: "height vs wavelength" shouldn't this be "wavelength vs height" (Y vs X)? - Figure3A caption: "estimates"
- Figure3C caption: ""single"
- Section 3.3, Line 11: "disequilibrium" and "single"
- Discussion line 30: is "(July)" missing after the 28.3%?

---

## Author Comment (AC2) · 8 Nov 2019

*We would first like to thank Robert Mahon for their thoughtful review. In the below document, the reviewers comments are in black; our responses to reviews are in blue italics.*

The authors present a systematic comparison of bedform bedload measurement techniques using a unique dataset. Using field data, as opposed to flume data as is often the case, the authors are able to investigate some of the complexities associated with systems evolving under unsteady flow conditions. The ultimate outcome of this paper can inform decisions on both multibeam sampling and processing strategies as well as the placement of single beam echosounder instrumentation on rivers to monitor bed- load flux. Thus the results of this paper are broadly relevant to river managers as well as to academic geomorphologists.

The overall flow and structure of the manuscript are quite clear. Figures are well placed into the manuscript context and are appropriate for fully describing the nature of the work. While I have no concerns that fundamentally call into question the nature of the science being done, there are a number of points which the authors could clarify or analyses that could be bolstered by more complete discussion. These comments are below:

I would like to see a description of the methods used to extract height and wavelength data from the BTT toolbox as it is a fundamental operation to the analysis in the paper. There are several methods for calculating these parameters, each of which have their respective advantages and disadvantages so it would be good for the authors to describe why the calculations employed in this toolkit are appropriate to their system.

*We have added the following to our description of the BTT: "After the BEPs are detrended, the BTT determines the zero upcrossing (i.e. points at which the profile positively crosses zero) and zero downcrossing (i.e. points at which the profile negatively crosses zero). The locations of crests and troughs are determined in the original BEP as follows: a crest is located at the maximum value between a zero up- and zero downcrossing; and vice versa, a trough is located at the minimum value between a zero down- and zero upcrossing. Bedform height is calculated at the vertical distance between crest and downstream trough. Bedform wavelength is calculated as the distance between two successive crests. For a more detailed explanation of the BTT please refer to van der Mark et al. (2008) and van der Mark et al. (2007)."*

Were bed elevation surveys corrected for apparent dilation as a function of the time between start and end of each multibeam survey? If not, was this considered and determined to be a negligible effect? See McElroy dissertation 2009, p. 44 (URI: http://hdl.handle.net/2152/1511).

*The average difference in time between surveys was around 10 minutes and departures from this were also order 10 minutes, therefore, while we acknowledge that the effect noted by the reviewer and McElroy (2009) is real, we determine it to be a negligible effect in the present study.*

A figure demonstrating the cross-correlation results would be good to show, as a lot of discussion is based on issues resulting from velocity calculations.

*See figure 3A.*

In Page 5 Line 6 the method for estimating wavelengths for the singlebeam experiment is described as the daily average from the repeat multibeam. I wonder if this introduces potential

for extra accuracy for this method that may not be possible in a situation in which a single beam fixed echosounder would be employed.

*Yes, this is most likely the case. Our single beam flux estimates are probably more accurate than what one might get using a different estimate of bedform wavelength.*

I would suggest more discussion of when a situation would arise where you have a measurement or a daily average of bedform wavelengths but only a single beam profile to estimate flux from. An alternative formulation might be to estimate wavelength using a height-wavelength relationship such as Bradley and Venditti, 2017 as this might be a more realistic representation of a likely application (i.e. a deployed single beam sensor established for continuous monitoring).

*Analyses of bedform fields throughout the Colorado river in Grand Canyon reveals that the Bradley and Venditti (2017) relations are a poor fit to observations. Because the bedform field is likely not in equilibrium with the flow due to daily fluctuations in discharge, i.e. because the dunes are always adjusting to flow, correlations between instantaneous bedform height or length and flow are not as robust as they otherwise would be. A better approach is to use mulitbeam measurements at multiple flows to develop a site-specific model for bedform dimensions as predicted by flow.*

What did the manual process entail for determining bedform velocity? Were you picking crests and tracking them? Looking at the slopes of the forms in the η(x,t) field (e.g. in Figure 2D)? It would be critical to determine whether the manual method itself includes any potential sources of bias in order to interpret its relation to the cross-correlation results.

*We picked crest locations and tracked them. We have added this information to the main text.*

I wonder if other methods for calculating bed velocity might be more appropriate than the cross-correlation method for this application, particularly given the unsteady flow conditions investigated. One example from Ganti et al., 2013 (doi:10.1002/jgrf.20094), their eq. 5 to compute the local velocity based on dividing the temporal change in local elevation by local slope at all points on the bed.

*The above mentioned method from Ganti et al. (2013) would most likely not apply to this data set. While computing local velocities is appropriate in flume experiments where the change in time between each successive bed elevation profile is 45 seconds, applying this method to field data with both a coarser spatial and temporal resolution would likely result in large errors (likely larger than those associated with the cross-correlation method).*

Were any physical bedload samples collected during the multibeam campaigns to compare with the ranges of flux measurements?

*No physical bedload samples were collected. We rather doubt the reliability of measurements from a bedload sampler lowered from a cableway suspended high above the water surface through 7m of the water column, into a field of dunes up to a meter high moving up to a meter per second.*

Some discussion is warranted of whether the bedform bedload equation of Simons et al., is even geometrically appropriate in situations where bedform growth/decay is occurring. I don't believe they considered this in their original work, and I am not aware of any later publications that show the validity of this method for non-steady bedform fields.

*Aside from the assumption that dunes are appropriately triangular, there is nothing in the Simons et al derivation that suggests it is not appropriate for application in time-discerete fashion such as here. That is to say, despite the growth and decay in dunes, the instantaneous bedform flux as predicted by their instantaneous geometry and celerity is appropriate and has been applied before to unsteady flows. While it is true that inferring wavelength from time-series of bed elevation measurements is made more difficult by spatially accelerating and decelerating dunes, the issue is a more in the implementation of the theory governing Simons et al, rather than the theory itself.*

Along similar lines I would encourage the authors to consider incorporating, or at least explaining the inappropriateness for their application, the insights from Guala et al. (2014, their Section 4 paragraph 2 in particular; doi: 10.1002/2013JF002759) in joint averaging of the elevation and velocity values.
*We agree the study of Guala et al is pertinent so we have added the following to the revised Discussion: "Guala et al. (2014) demonstrate a frequency dispersion in the relationship between dune celerity, $V_c$, and wavelength, $\lambda$, because small dunes tend to move faster than larger ones. This doesn't bias our computed bedload fluxes from multibeam data since we use time-series of bedform statistics from $\eta(x,y,t)$, however it does place limits on any calculation of equivalent statistics from $\eta(t)$ because it requires assuming a model that relates average $V_c$ with average $\lambda$, or rather that the functional form between them doesn't vary in time, which may not be strictly true."*

While somewhat outside the scope of the review of the paper itself, I should note that the license type given to the dataset and code hosted in the SEAD repository is potentially quite restrictive to some river management uses and researchers, given that it does not allow commercial use or any derivatives. This may be less important for the data itself, but it may heavily limit the use of this work to have code that cannot be modified. A share alike restriction, for example, would make this more accessible.
*We have updated the license to allow for commercial use and derivatives.*

Line Comments: The following line specific comments are non-critical to the science of the manuscript and are meant to help improve readability or clarity.
*We have corrected the below Line Comments in the manuscript.*

Page 1 Line 2: "remains elusive" is relatively non-concrete and feels dismissive of the wealth of literature and practice on field-scale bedload measurement techniques spanning half a century or more.
Page 1 Line 14: references are missing at "(e.g. ?)"
Page 1 Line 20: References such as Simons et al., 1965 and others don't explicitly derive from the exner equation, per se. They are derivations of mass conservation but not necessarily predicated on Exner's formulations.

Page 2 Line 1: Simons wasn't the first to show this, as written. For example, Bagnold 1941, Chapter 13 derives a similar formulation, albeit with some geometric inaccuracies. I suggest simply removing the word "first" from the sentence.

Page 2 Line 9: remove comma after ". . .discharge conditions,"

Page 2 Line 12: is there a reference for ". . .bedload flux estimated from translating dunes remains one of the most accurate. . ."?

*This claim has previously been made by Wilbers and Ten Brink (2003) and Nittrouer et al. (2008) who said the measurements came with " ... relatively high accuracy so long as the dune geometry and translation distances are large relative to the positioning error" which is the case here. The relative disadvantages of direct sampling are well documented (e.g. Holmes and Holmes, 2010), for example direct sampling disturbs the flow and therefore the rate of bedload, and the sampler must be placed squarely on the bed surface to adequately sample, therefore the presence of dunes causes error.*

Page 4 Line 14: ISDOTTv2 is not a familiar/common tool since it is not public. If you wish to include this statement, it would be good to describe what that tool is and why it would be useful here. Otherwise I would suggest removing it.
*We have removed this statement.*

Page 4 Line 16: please describe the "missing triangles" correction.

Page 4 Line 22: consider rewording as it states a 1965 reference is based on a 2005 reference.

Page 7 Line 14: ". . .for growing (shrinking) dunes is 1.2 (0.75)." I suggest rewording to ". . .for growing and shrinking dunes is 1.2 and 0.75, respectively."

Figures: for figures 1, 2 and 4 there are abbreviations used which would be helpful to have defined in figure captions so the reader doesn't have to remember or find from the text. BEP, RMB, SB and MSB are all used. Additionally, Xcorr and RMSE are used but not defined in captions or in the text body.

---

## Author Comment (AC3) · 8 Nov 2019

*We would first like to thank Kory Konsoer for their thoughtful review. In the below document, the reviewers comments are in black; our responses are in blue italics.*

In "Estimating Sand Bedload in Rivers by Tracking Dunes: a comparison of methods based on bed elevation time-series", the authors present a systematic comparison for different approaches for estimating bedload transport based on dune migration. The methods compared rely on repeat multibeam echo sounding surveys from a reach of the Colorado River during two different field campaigns that exhibit different discharges. The multibeam surveys provide the base data, and three different subsets from the data are selected. The three datasets used in the comparison are, 1) longitudinal transects of bed elevation from the full multibeam surveys, which provide spatial data series, 2) extraction of bed elevation at a single point over time (temporal), and 3) extraction of bed elevation at multiple points over time (temporal). The authors also include synthetic sinusoidal signals that are used to evaluate bedform dynamic of growing/shrinking size that would occur during unsteady flows.

Overall the paper is well written and organized, and the presentation of the results is very clear. The topic of this paper is also of great importance as river scientists still struggle with determining best practices for quantifying bedload transport rates. However, I would recommend addressing a few issues related to the methods and discussion before the manuscript should be accepted for final publication. I outline these below.

Although the data are measured using a multibeam echo sounder, the dataset is not fully utilized and instead only bed elevation profiles are extracted. Thus, the comparisons are essentially spatial series of single beam, stationary single beam, and stationary multi-single beam. It is stated that the reason for this is to account for anisotropy among the different methods equally (page 4, lines 11-14), which is understandable. However, as is stated more than twice throughout the manuscript, multibeam surveys are considered the most accurate due to the high spatiotemporal resolution, yet are not being used to their full potential.

Why have you decided not to include the full three- dimensionality of the multibeam survey when considering sediment transport? If you consider this to be most accurate, then you could conceivably have a fourth method using the repeat multibeam surveys as two dimensional differencing compared to the three "single beam" methods presented in the paper. *This is essentially the ISDOT method (Abraham et al., 2011), which requires there to be conservation of mass over the survey area (all sediment eroded from the area is deposited in the same area). Additionally, this method is designed for bedforms moving at a constant speed, with little to no deformation, and little to no suspended sediment. In our field data, the dunes change speed throughout the day, change shape significantly, and suspended sediment is available. Although the ISDOT method works well in a flume setting, we don't feel that it is applicable to our field data.*

Similarly, it appears as though all the repeat multibeam bed elevation profiles have been averaged into a single value for the area of interest. Why not keep these separate and evaluate the comparisons spatially? *The repeat multibeam profiles are only averaged at each location, so within the area of interest there are 40 daily bedload transport estimates. We compute a daily average at each location because it is directly comparable to the measurements made by single beam and multiple-single beam echosounders. The CDFs in this paper illustrate the distribution of daily average bedload transport estimates for the entire area of interest.*

From the bed elevation raster shown in figure 1 there appears to be quite a difference in elevation and bedform size from the left bank (higher bed elevation) to right bank (lower bed elevation). Is there a systematic difference in the comparisons from left to right? If so, is it related to bedform dimensions? *We have added a section to the results to address this point. Please see section 3.5 in the updated manuscript.*

This spatial information would be extremely relevant for the discussion section. In particular, one of the topics I felt was missing from the discussion was how the findings of this study can be used to provide insight on where stationary single beam sensors could be installed. My understanding is that most single beam sonars are attached to bridge piers or off banks/docks. If a spatial component of comparison is included in this paper, it would be possible to inform deployments in future studies. Do your comparisons show less agreement between the methods closer to the bank? These are questions easily answered from your dataset without much additional analyses. *We have added a paragraph to the discussion section to address this point. Please see page 10, line X6*

Could you provide more information on how the cumulative density plots are prepared? It is stated on page 4 line 30 that Eq. 1 is averaged over a dune field. There is no mention of how the CDF are prepared. How many bed elevation profiles are needed before a 'stationary' average bedload transport rate is obtained? How far apart do the lines need to be? Answers to these questions could help guide surveys using boat- mounted single beam sonars. (it is stated that this is not of concern for the paper, however the extracted profiles from the multibeam survey is essential that). *Equation 1 produces a bedload transport estimate that is the average for the entire bed elevation profile. This is because we are using an average bedform height and average dune celerity. Therefore, each timestep at each location has one bedload transport estimate. We then average all timesteps at each location for a daily average bedload transport rate. Thus CDFs for*

*repeat multibeam July data contain 20 estimates of daily bedload transport while repeat multibeam CDFs for March contain 40 daily bedload transport estimate. Single beam CDFs contain 20 and 40 bedload transport estimates for July and March data respectively.*

There is reference to a figure 5 and figure 6 on page 6, but figures are only 1-4. I have attached an annotated pdf with other technical issues. Please see for grammar and other comments.
*Thank you and apologies for the confusion. Those figure references were for a previous version and were mistakenly left in this version.*

Please also note the supplement to this comment: https://www.earth-surf-dynam-discuss.net/esurf-2019-38/esurf-2019-38-RC3- supplement.pdf
*We have corrected the grammatical and spelling errors highlighted in this supplement.*